# Predicting Factor for Occurrence of Postoperative Pancreatic Fistula in Patients with Pancreatic Neuroendocrine Tumors

**DOI:** 10.3390/diagnostics15030268

**Published:** 2025-01-23

**Authors:** Nutu Vlad, Florina Delia Andriesi-Rusu, Andrei Chicos, Ana Maria Trofin, Ramona Cadar, Mihai Lucian Zabara, Delia Ciobanu, Mircea Costache, Corina Lupascu-Ursulescu, Alin Mihai Vasilescu, Costel Bradea, Mihaela Blaj, Oana Maria Lovin, Adi Ionut Ciumanghel, Felicia Crumpei, Cristian Dumitru Lupascu

**Affiliations:** 1First Surgical Clinic, “St. Spiridon” Hospital Iasi, Independentei str, no 1, 700111 Iasi, Romania; nutu.vlad@umfiasi.ro (N.V.); andrei.f.chicos@umfiasi.ro (A.C.); ana-maria.trofin@umfiasi.ro (A.M.T.); mihai-lucian.zabara@umfiasi.ro (M.L.Z.); costache.mircea@gmail.com (M.C.); costel.bradea@umfiasi.ro (C.B.); felicia_crumpei@yahoo.com (F.C.); cristian.lupascu@umfiasi.ro (C.D.L.); 2Faculty of Medicine, ”Grigore T. Popa” University of Medicine and Pharmacy Iasi, 700115 Iasi, Romania; delia.ciobanu@umfiasi.ro (D.C.); corina.ursulescu@umfiasi.ro (C.L.-U.); mihaela.blaj@umfiasi.ro (M.B.); oanamarialovin@gmail.com (O.M.L.); adi.ionut80@yahoo.com (A.I.C.)

**Keywords:** pancreatic neuroendocrine tumors, pancreatic neuroendocrine carcinomas, pancreatic resections, postoperative pancreatic fistula, prognostic factors

## Abstract

**Background:** Neuroendocrine tumors are tumors that can develop in any organ but show a predilection for the pancreas. These can be secreting or non-secreting tumors, or they can be well differentiated or poorly differentiated, or neuroendocrine carcinomas. Surgical treatment is the only treatment with curative intent, but postoperatively, it shows an increased incidence of postoperative pancreatic fistulas. **Methods:** We carried out a retrospective study which included 26 patients with neuroendocrine tumors and neuroendocrine carcinomas, for whom we performed cephalic duodenopancreatectomies, distal pancreatic resections or enucleation. **Results:** In our study group, the incidence of pancreatic fistulas was 28%, and a series of risk factors such as the type of surgery (duodenopancreatectomy and enucleation were associated with the highest incidence of POPF), histological type (pancreatic neuroendocrine carcinomas were associated with lowest incidence of POPF), obesity (the incidence of POPF was double in the obese group), functioning tumors (with *p* = 0.032 and AUC = 746) and dynamic hemoglobin value (AUC = 705 shows a good predicting power, with a cutoff value = 1.8 drop hemoglobin) were indicated. **Conclusions:** Neuroendocrine tumors show a predisposition for the occurrence of postoperative complications, especially postoperative pancreatic fistulas. There are multiple risk factors that interact in the production of postoperative complications.

## 1. Introduction

Neuroendocrine neoplasms (NENs) represent a group of tumors which have their origin in neuroendocrine cells, with a variable biologic behavior and wide heterogeneity [1,2,3]. They can develop in any organ, having a predilection for the pancreas, lungs and digestive tract [1,4]. According to World Health Organization Classification, 2017 and 2019 editions, NENs can be classified into well-differentiated G1, G2 and G3 neuroendocrine tumors (pNETs) or poorly differentiated G3 tumors (pancreatic neuroendocrine carcinomas, pNECs), with it mentioned that all pNETs have malignant potential [1,4,5].

pNETs were first described in 1869 as a distinct tumor with distinct biological behavior from pancreatic adenocarcinomas [1]. pNETs can secrete different types of hormones; depending on this characteristic, they are classified into functional tumors and non-functional tumors [1,6]. Patients with functional pNETs are symptomatic in a proportion of up to 90%, the symptomatology being given by the hormonal hypersecretion of the tumor (insulin, serotonin, gastrin, glucagon, vasoactive intestinal peptides, somatostatin) [1,4,6].

Surgical resection is the only curative treatment for patients with pNETs (localized tumors), with a 5-year survival of more than 90%, while for advanced tumors or metastatic disease, systemic therapy with somatostatin analogs and chemotherapy is recommended [1,7,8,9]. At the time of diagnosis, approximately 50% of cases have metastatic disease, and 20% have locally advanced tumors [10]. Surgical resections are indicated especially for functioning pNETs, regardless of tumor size, to control hormone hypersecretion, or when there are local compressive symptoms, and for non-functioning pNETs greater than 2 cm [3,11,12]. For patients with unresectable tumors, survival at 5 years varies between 15 and 27% [7].

Pancreatic neuroendocrine tumors are associated with an increased incidence of pancreatic fistulas after pancreatic resections [13,14]. A pancreatic fistula is the most frequent and severe postoperative complication, after pancreatic resections, with an important impact on quality of life, and an increased risk of bleeding, abscess, sepsis and multiple organ failure [11,12].

In 2016, ISGPS (International Study Group for Pancreatic Surgery) introduced new criteria for determining the severity of POPFs (postoperative pancreatic fistulas), and the major change was the replacement of grade A fistulas with the term “biochemical leak” (asymptomatic pancreatic leak). Grade B included cases with invasive procedures for the drainage of abdominal collections and angiographic procedures and grade C included patients with organ failure, with surgical reinterventions or death, both of them being included in the CR-POPFs (clinically relevant postoperative pancreatic fistulas) [15].

## 2. Material and Methods

Our retrospective study includes a selection of cases over a period of 10 years, with pancreatic resections for neuroendocrine tumors. Initially, patients with pancreatic resections and enucleation were selected; then, the histological reports were analyzed and only patients confirmed with neuroendocrine tumors were included in the study. Preoperatively, all patients were evaluated through paraclinical investigation and imaging (abdominal ultrasound, computer tomography), thus establishing the diagnosis of pancreatic tumor. No preoperative tumor biopsies or EUS-guided fine needle aspirations were performed. The diagnosis of pancreatic neuroendocrine tumor was established histologically, after performing immunohistochemical staining (IHC) that was positive for synaptophysin, chromogranin A and the Ki67 mitotic index (Figure 1). In this study, we aimed to follow the postoperative complications and the identification of risk factors, and the prognostic factors for postoperative survival.

The diagnosis of pancreatic fistula was established according to the ISGPF classification (International study group for pancreatic fistula) when there was a persistence of drainage and an increase in amylase values in the liquid on the drain tube.

The database included general information (age, gender), information related to the type of surgical intervention, histology and tumor staging, the dynamics of some biological factors (proteins, hemoglobin, liver balance), postoperative evolution, postoperative complications, comorbidities and survival.

The SPSS program was used for statistical analysis, and the statistical association was considered at a *p*-value lower than 0.05. Continuous variables were reported as mean with standard deviation. Comparisons between the analyzed groups were performed using the t-Student test, ANOVA, Kruskal–Wallis or Mann–Whitney U Test for continuous variables. The homogeneity of the series was verified regarding the statistical differences between the variances of the series by the Levene test (Levene Test of Homogeneity of Variances). The correlations between certain parameters were tested using the Pearson test, by evaluating the correlation coefficient r, and the Spearman’s rank correlation coefficient or Kendall’s tau correlation. Qualitative variables were presented as absolute (*n*) and relative (%) frequencies, and comparisons between groups were made based on the results of non-parametric M-L, Yates or Pearson Chi-square tests. The power of univariate prediction of risk factors was assessed using the ROC curve based on the value of the area under the curve (Area Under the Curve: AUC).

## 3. Results

### 3.1. Characteristics of Study Group

The study performed is a retrospective study, which included a group of 26 patients, consisting of 16 women and 10 men, aged between 31 and 81 years, with a median of 55 years.

The size of the tumor varied between 0.9 cm and 6.5 cm, with a median of 2 cm, and peaked in the range of 1–1.5 cm, according to the histogram in Figure 2.

### 3.2. Histologically and Functioning Characteristics of Tumors

From the point of view of hormonal secretion, 9 patients had functional tumors (7 with functioning pNETs and 2 with functioning pNECs) and 17 had non-functional ones (12 with non-functioning pNETs and 5 with non-functioning pNECs).

Among the patients with functional pNETs, five had insulinomas and two had gastrinomas, with all patients presenting symptoms specific to hormonal hypersecretion. Both patients diagnosed with gastrinomas also had associated MEN1 syndrome (multiple endocrine neoplasia type1—a rare inherited cancer syndrome, characterized by the development of multiple neuroendocrine tumors of the parathyroids, gastro-entero-pancreatic tract and anterior pituitary gland, and less commonly, the adrenal cortical gland, thymus and bronchi). The association of MEN1 syndrome was also identified in a case with non-functional pNET. (Table 1)

In our study group, the incidence of pNECs was 26.92% (7 cases). In two cases, pNEC was identified in patients with functional pNETs (one case with insulinoma-like neuroendocrine carcinoma, and one case with gastrinoma associated with MEN1 that developed postoperative distant metastases), the rest being in the group of patients with non-functional pNETs. One case had a MANEC tumor type (mixed adenoneuroendocrine carcinoma) (Table 1).

**Table 1 diagnostics-15-00268-t001:** Characteristics of tumors.

	pNETs	pNEC	Total
Non-functioning tumors	12	5	17
Functioning tumors	7 (5 insulinomas and 2 gastrinomas)	2 (1 case with insulinoma-like neuroendocrine carcinoma and 1 case with malignant gastrinomas who developed metastasis postoperatively)	9
total	19	7	26

Additionally, there were cases of pNETs that were associated with other primary malignant tumors. One of the patients, 7 years after the enucleation of an insulinoma, was diagnosed with locally advanced ovarian tumor with invasion of the rectosigmoid, ureter and bladder, with peritoneal carcinomatosis, and the tumor biopsy confirmed the diagnosis of poorly differentiated ovarian carcinoma with a high mitotic index. Another case was a tumor collision between a gastrinoma and pancreatic acinar carcinoma. A case with a multicentric pNEC, for which a total duodenopancreatectomy was performed, developed an invasive ductal breast carcinoma one year after pancreatic surgery and a bronchopulmonary cancer after 3 years.

### 3.3. Surgical Type

In regard to the type of surgical intervention, distal pancreatectomies (distal pancreatic resections with or without preservation of the spleen) prevailed in 12 cases (46.15%), followed by duodenopancreatectomies Whipple type in 8 cases (30.77%). Enucleations were also performed in five cases, and a total duodenopancreatectomy in one case.

In one case, the duodenopancreatectomy and the resection of liver metastasis were performed at the same operative time, for a young patient of 31 years old.

### 3.4. Postoperative Complications:Pancreatic Fistulas

The incidence of postoperative pancreatic fistulas was 28%, after the case of total duodenopancreatectomy was excluded from the statistical analysis. The highest incidence of POPF was observed in the group of patients with duodenopancreatectomy (50%), followed by enucleation (40%), and the lowest in the group of patients with distal pancreatectomy (8.3%) (Table 2).

Most of the fistulas were type B, with two cases type A after enucleation and a single case with POPF type C after a duodenopancreatectomy, which required surgical reintervention.

**Table 2 diagnostics-15-00268-t002:** Surgical interventions versus postoperative pancreatic fistula.

		POPF		Total
		No	Yes	
Duodenopancreatectomy (Whipple type)	N	4	4	8
	%	50.0%	50.0%	100.0%
Distal pancreatic resections	N	11	1	12
	%	91.7%	8.3%	100.0%
Enucleation	N	3	2	5
	%	60.0%	40.0%	100.0%
Total	N	18	7	25
	%	72.0%	28.0%	100.0%

### 3.5. Association Between Obesity and POPF

The incidence of obesity in our study group was 42.31% (11 cases). Although we did not obtain a significant statistical association (*p* = 0.261), 57.1% of the patients who developed pancreatic fistula postoperatively had different degrees of obesity.

In the group of obese patients, the incidence of pancreatic fistula was double, 40% versus 20%, respectively, that in the group of non-obese patients (Figure 3).

The aspect of the ROC curve (Figure 4) and the calculated value under the curve of AUC = 0.619 reveal the fact that the presence of obesity has a good power of prediction for the risk of postoperative pancreatic fistula.

### 3.6. Association Between Tumor Size and POPF

In our study group, we found no statistically significant association between tumor size and the risk of postoperative pancreatic fistula (*p* = 0.384), with a median of 2 cm in the group with pancreatic fistula vs. 2.2 cm in the group without pancreatic fistula. However, 57.1% of patients who developed postoperative pancreatic fistula were in the group of patients with tumors smaller than 2 cm (Table 3). The association between the development of POPF and tumors under 2 cm can be correlated with the fact that enucleation was chosen for small tumors, if the conditions for this technique were met.

### 3.7. Association Between Functioning Character and POPF

The statistical analysis showed an important statistical association between the functioning character of the tumor and the risk of pancreatic fistula (*p* = 0.032), with the incidence of pancreatic fistula being 71.4% in the group of functioning tumors.

The aspect of the ROC curve (Figure 5) and the calculated value of AUC = 0.746 reveal the fact that the functioning character of the tumor has a high predictive power for the risk of postoperative pancreatic fistula.

These results are also supported by correlation tests such as Spearman’s rank correlation coefficient or Kendall’s tau correlation, with a *p* = 0.012 (Table 4).

### 3.8. Association Between Malignant Tumor and POPF

Although a statistically significant association was not obtained, the lowest incidence of postoperative pancreatic fistulas was in the group of patients with malignant tumors (16.67%) (Figure 6). Approximately one-third of pNET patients developed POPF (31.58%) (Figure 6). The incidence of POPF in the group of patients with pNETs was almost double that in the group of patients with pNECs.

Of all pNEC patients, one-third presented with tumors with a diameter smaller than 2 cm, this aspect calling into question the “watch and wait” recommendation for small tumors.

A total of 71% of the cases of tumors with malignant behavior presented positive lymph nodes, and of these, 66% had tumors smaller than 2.2 cm in size.

### 3.9. Prophylactic Sandostatin Administration Versus Occurrence of POPF

Depending on the preferences of the main operator, a series of patients received prophylactic somatostatin in the postoperative period. The incidence of POPF in the group of patients who received prophylactic sandostatin was 50%, versus 7.69% in the group of patients without prophylactic sandostatin (Figure 7). Of the total number of patients who developed postoperative POPFs, 83.33% were cases that received prophylactic sandostatin (Figure 8).

These results are also supported by correlation tests such as Spearman’s rank correlation coefficient or Kendall’s tau correlation, with a significative statistic of *p* = 0.013 (Table 5).

### 3.10. Hemoglobin Value Variation Versus POPF

Another analyzed factor was the dynamic value of hemoglobin. In our study group, neither the preoperative hemoglobin value nor the hemoglobin value on the first postoperative day were correlated with an increase in the risk of pancreatic fistula. But the dynamic analysis of hemoglobin was identified as an important predictive factor, so a good statistical correlation was obtained between the decrease in hemoglobin from the first operative day compared to the initial value and the increase in the incidence of pancreatic fistula (which means that a drop in hemoglobin value is associated with an increase in the incidence of pancreatic fistula).

The AUC = 705 value confirms a good predictive power of the drop in hemoglobin value for the risk of pancreatic fistula. Thus, a cutoff value of 1.8 was calculated, with a sensitivity of 67% and a specificity of 37% (Figure 9).

The subsequent analysis according to the cutoff value of the hemoglobin drop showed a 2.2 times higher incidence of POPF in the group of patients with a hemoglobin drop >1.8 (Figure 10).

Moreover, two-thirds of the patients with a POPF had a hemoglobin drop greater than the calculated cutoff value (Figure 11).

Statistical analysis showed that there is a strong correlation between the hemoglobin value drop and the occurrence of postoperative pancreatic fistulas, with a *p* = 0.44 (Table 6).

### 3.11. Survival

The 2-year survival rate of the group was 80.77%, with only one case of in-hospital death. Postoperative survival varied between 13 months and 102 months, with a median of 47.24 months.

## 4. Discussions

Pancreatic neuroendocrine tumors are rare tumors, derived from pancreatic neuroendocrine cells, representing about 3–5% of all pancreatic tumors [2,16,17]. They are the second most common epithelial malignancy of the pancreas [18].

The immunohistochemical profile is essential for establishing the diagnosis of pNETs, this being confirmed by the presence of synaptophysin, chromogranin A (CgA) and Ki67 expressions [4]. Chromogranin A is a glycoprotein secreted by neurons and neuroendocrine cells; therefore, it represents a diagnostic marker, but at the same time, serum CgA is an important marker in postoperative monitoring for non-functional pNETs, being secreted by 60–100% of non-functioning pNETs [1,7]. CgA is considered an independent prognostic factor for survival from and recurrent pNETs [1,7]. The disadvantage of CgA is that it can be false positive in chronic gastritis, inflammatory bowel disease, pancreatitis, renal or liver failure, other malignancies (thyroid and prostate cancer) or different therapies (proton pump inhibitors, somatostatin analogues or steroids) [1,7,18]. All patients included in our study were monitored postoperatively by the endocrinology service, and dynamic dosages of chromogranin were performed (elevated values were suggestive of postoperative distant metastases).

In two of the cases with neuroendocrine carcinomas, the dosage of chromogranin A showed that an increase in CgA values was suggestive and was associated with the imaging appearance of metastases.

The Ki67 mitotic index is an important marker both for prognosis and for establishing the therapeutic strategy [4]. The pNETs in G3 with Ki67 > 20% have a poorer median survival (54.1 months) than pNETs in G1 (Ki67 < 2%) and G2 (Ki67 < 20%) with a median survival of 67.8 months [17].

In our study group, it was not possible to analyze the survival according to the mitotic index ki67 because the survival rate at 2 years was over 80%

To differentiate a G3 pNET from a pNEC, staining for p53 and RB1 is required [4].

A new concept was introduced that defines a mixed neoplasm that includes both a neuroendocrine component and a non-neuroendocrine component—MANECs (mixed adenoneuroendocrine carcinomas) [1]. In our study group, also, there was a case with a MANEC-type tumor.

Insulinomas are the most frequent functioning pNETs with, generally, a tumor size under 2 cm, and they show benign behavior, unlike the other functional pNETs that show malignant behavior [4].

In our study, there was a case with an insulinoma-like neuroendocrine carcinoma, which had a remission of hypoglycemia episodes in the postoperative period, was monitored and 6 years postoperatively developed distant metastases. Malignant insulinomas are rare tumors accounting for approximately 20% of all insulinomas. In most cases, the patient becomes symptomatic after the development of liver metastases when there are enough insulin-secreting cells to trigger the hypoglycemic syndrome [4].

Approximately 30% of pNETs are gastrinomas, which due to the excess of secreted gastrin cause peptic ulcers (sdr Zollinger-Ellison) [18]. Gastrinomas can have multiple locations, with pancreatic ones often being associated with duodenal gastrinomas [18]. When a gastrinoma is associated with MEN1, there is extremely rarely a single lesion; generally, there are multiple pancreatic or gastrointestinal locations [7].

One of our patients included in the study also had gastrinomas with multiple locations: pancreas, duodenum and stomach, and these were associated with MEN1 syndrome. In this case, multiorgan resection was mandatory. In this case, the postoperative evolution was simple, without complications.

Up to 10% of pNETs can be associated with other genetic syndromes (most frequently with MEN1) or other multiple primary malignancies (MPMs) [2,4,9,18]. Surgical treatment is recommended for patients with MEN1 associated with functional pNETs but contraindicated when MEN1 is associated with non-functional pNETs [18]. pNECs are rare, solitary tumors and are not associated with genetic syndromes [4]. Over 80% of patients with MEN1 are associated with pNETs, and surgical treatment is required to prevent malignant progression when the endocrine syndrome is refractory to treatment, or for tumors larger than 2 cm or showing fast growth [19].

Gastrinomas are malignant in a proportion of 60–90%, and the presence of metastases represents the most reliable predictive factor of long-term survival [7]. Considering the high rate of nodal metastases, the recommended surgical treatment in the case of gastrinomas is represented by pancreatic resections associated with regional lymphodissection [7]. Intraoperatively, it is mandatory to carefully explore the adjacent organs so that other extrapancreatic gastrinomas can be identified [7]. For this purpose, digital palpation, intraoperative ultrasound, duodenotomy or endoscopy with transillumination can be performed [7].

Generally, non-functional pNETs are diagnosed when they reach large sizes [9,18]. This aspect can be debatable between the hypothesis that non-functional tumors have a fast growth rate, or the lack of symptoms leads to a late diagnosis.

While Wang et al. showed that lymph node status alone was not a significant predictor of survival in univariate and multivariate analysis [5], other studies concluded that lymph node metastasis is an important prognosis factor [2]. In addition, NCCN guidelines recommend lymph node dissection for all functional pNETs, regardless of tumor size [2].

Except for patients with enucleation, pancreatic resections were accompanied by lymph node dissection.

The occurrence of postoperative complications is also a prognostic factor, with the most common complication being pancreatic fistulas, which are associated with multiple comorbidities and have an important impact on postoperative evolution and survival [11,14]. Despite the high rate of postoperative complications, surgical treatment remains the only treatment with curative potential both for tumors with malignant behavior and for the control of hormonal hypersecretion [11].

In our study, it was not possible to analyze the impact of POPFs on survival because survival at 2 years was over 80%, and the small study group did not allow us to calculate statistical indicators.

There are studies that show that in neuroendocrine tumors there is no desmoplastic stroma (which is well represented in pancreatic adenocarcinomas) or fibrosis, with this desmoplastic stroma being a protective factor for the appearance of pancreatic fistulas [6,14]. Additionally, Hedges et al. observed in their study, an increased incidence of POPF in the group of patients with pNETs, compared to the non-pNET group, thus supporting the importance of the histological type as a risk factor for the development of POPF [12].

For the entire study group, the incidence of pancreatic fistula was 28%, but an increased incidence was observed especially after pancreaticoduodenectomy and enucleation (50% and 40%, respectively). Inchauste et al., in a study that included 122 patients with pNETs, observed an increase in the incidence of postoperative pancreatic fistula in patients with enucleation versus pancreatic resections [11]. The same study identifies obesity as an important risk factor for postoperative pancreatic fistulas, especially for cases with pancreatic resections, but not for enucleation [11].

In our study, the incidence of postoperative pancreatic fistula was also identified as being double in the group of obese patients compared to the group of non-obese patients. Moreover, advanced statistical analysis, such as the ROC curve, confirmed the fact that obesity is an important predictive factor for postoperative complications.

The presence of obesity predisposes to complications postoperatively such as hemorrhages, abdominal wall infections and anastomotic fistulas, with direct effects on the increase in hospitalization days and hospital costs. The association of obesity with pancreatic steatosis and large intraoperative blood losses represents an important risk for the occurrence of pancreatic fistulas, especially in those of type C, which can be responsible for up to 41% of all postoperative deaths [6,13,20].

There are studies which determined that visceral fat is a more accurate predictor of postoperative pancreatic fistulas than body mass index, for cases of pancreatic resections [6,13]. For enucleation cases, the most important risk factor for the occurrence of POPFs is the location of the tumor in the proximity of the pancreatic duct (at a distance of less than 2 mm); in these cases, the anatomical conditions do not allow for limiting the occurrence of a pancreatic fistula [13].

Generally, enucleation is indicated for tumors smaller than 2 cm or for peripherally located insulinomas [7]. The criteria that guide the enucleation option refer to the size of the tumor, non-malignant lesion and proximity to the pancreatic ducts [21]. The advantages of enucleation are reduced intraoperative blood loss, shorter operating times and better preservation of pancreatic tissue [6,21]. Even if a higher incidence of pancreatic fistulas was observed after enucleation, these were less severe than after pancreatic resections [21]. In our study, the incidence of pancreatic fistulas was much higher (by 40%) in the group of patients with enucleation, according to other studies. Atema et al., in their study, obtained a strong correlation between the type of surgical intervention and the risk of POPF, with an increased incidence especially after enucleation or central pancreatectomy [6]. However, in our study group, despite the increased incidence, the pancreatic fistulas that developed after enucleation were pancreatic fistulas with low debit, type A, and the evolution was slowly favorable and without major implications on the postoperative evolution.

For small tumors with low malignancy potential, enucleation remains a viable and safe option, with studies showing long-term survival comparable to pancreatic resections [21].

Another important factor in the occurrence of POPFs is intraoperative blood loss, considering that a loss of more than 1000 mL is associated with a very high risk [14].

In our study group, we observed a strong statistical association between the decrease in hemoglobin in dynamics on the first postoperative day compared to the preoperative value of hemoglobin. Statistical analysis showed that a drop in hemoglobin greater than 1.8 mg/dl was associated with a doubling in the incidence of POPF. This aspect leads to the conclusion that the body compensates less for a rapid decrease in hemoglobin. The factors that can lead to a decrease in hemoglobin can be intraoperative blood loss, or the administration of large amounts of fluids intraoperatively or in the preoperative period. In this study, we observed that patients tolerated well a progressive decrease in hemoglobin (low hemoglobin values at admission did not represent a risk factor for POPF), in contrast to the decrease in hemoglobin immediately postoperatively, for which a strong association was obtained through statistics.

Molasy et al. observed in their study, a 5-times higher incidence of POPF in the group of patients who had an intraoperative blood loss greater than 700 mL [14].

For tumors larger than 2 cm, surgical treatment is unanimously accepted, regardless of the tumor stage and grade, due to the risk of distant metastases [22]. However, a topic that still incites debate is the indicated treatment in non-secreting tumors smaller than 2 cm. Pitt et al. draw attention to the fact that the risk of malignancy is also present in the case of small tumors; in their study, 4% of patients with tumors smaller than 3 cm, without metastases and negative nodes, presented recurrence in the postoperative period [21].

Additionally, in our study, one-third of the patients with a pNEC had a tumor size smaller than 2 cm, which supports the indication of surgical treatment even for small tumors, at the detriment of surveillance and monitoring. Studies performed on cases of small pNETs obtained a significantly higher 5-year survival in cases with pancreatic resections versus non-surgical ones [2]. In this context, we consider that surgical treatment is the first option even for small tumors due to the risk of malignancy.

Opting for surgical treatment must also consider the rate of postoperative complications, so the ENET guidelines (European Neuroendocrine Tumors Society) recommend surgical treatment for small lesions when they show a growth rate greater than 0.5 cm over a period of 6–12 months [7].

There are studies that have shown that up to 26% of non-functional pNETs between 1 and 2 cm in size have positive lymph nodes, and those under 1 cm have positive lymph nodes in proportion to 12% [2,7]. In our study, 66% of the patients with lymph node metastases were cases with tumors smaller than 2.2 cm. In this context, the question arises whether enucleation is an option for non-functional, small pNETs, in conditions where lymph dissection cannot be performed. Lymph node metastasis is an important factor, both for staging (positivity classifying pNETs in stage III) and as a postoperative prognostic factor [2].

For patients with liver metastases at the time of diagnosis, surgical treatment is indicated if two conditions can be met: the affected liver tissue is below 50% of the total volume and it is possible to resect at least 90% of the tumoral tissue [1,9]. Even if the recurrence rate for these patients is high, it has been observed that in the long term, an increase in survival is obtained [1,9].

In one of the cases in our study, with liver metastasis at the time of diagnosis, a duodenopancreatectomy operation associated with the resection of the liver metastasis was performed at the same time, in a 31-year-old patient, with a good postoperative survival (he is still being monitored after 43 months postoperatively).

Contrary to these results, the guidelines of the NANETS (North American Neuroendocrine Tumor Society) do not mention surgical treatment as an option for metastatic pNEC [16]. On the other hand, the resection of the primary tumor can improve the prognosis in metastatic pNETs, and a longer survival [2,16].

## 5. Conclusions

Neuroendocrine tumors are a group of heterogeneous tumors, with a varied histological and immunohistochemical profile. These tumors show a predisposition for the occurrence of postoperative complications, especially postoperative pancreatic fistulas. Several additional factors, such as the type of surgical intervention, obesity, administered medication and dynamic variation in hemoglobin, are factors that increase the risk of postoperative complications.

## Figures and Tables

**Figure 1 diagnostics-15-00268-f001:**
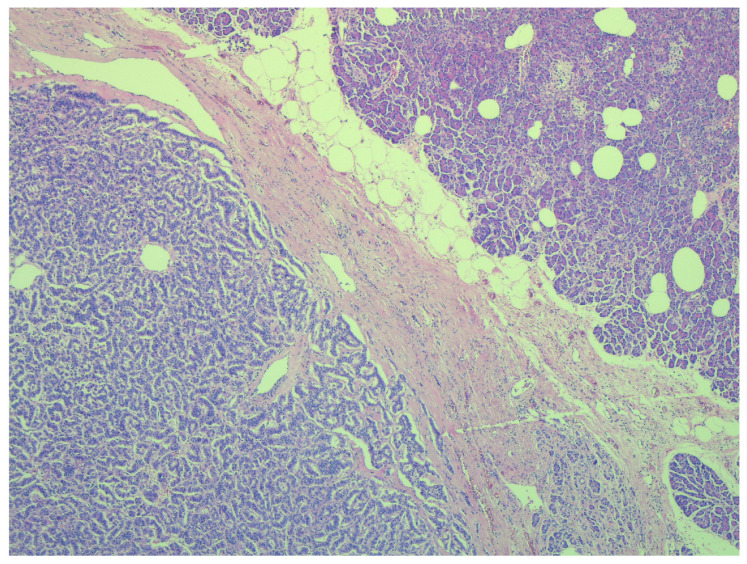
pNET—Hematoxylin–Eosin staining.

**Figure 2 diagnostics-15-00268-f002:**
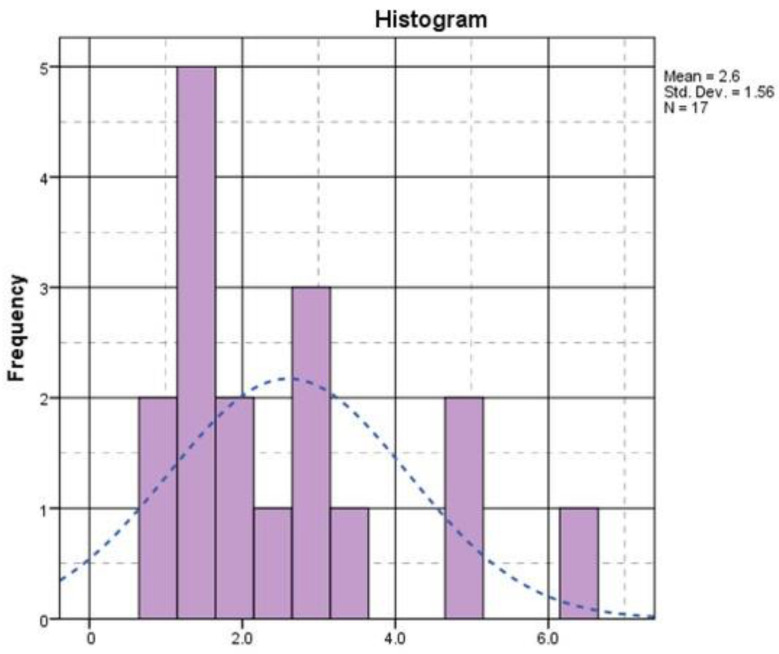
Histogram of tumor size diameter (the x axis = the tumor size in cm).

**Figure 3 diagnostics-15-00268-f003:**
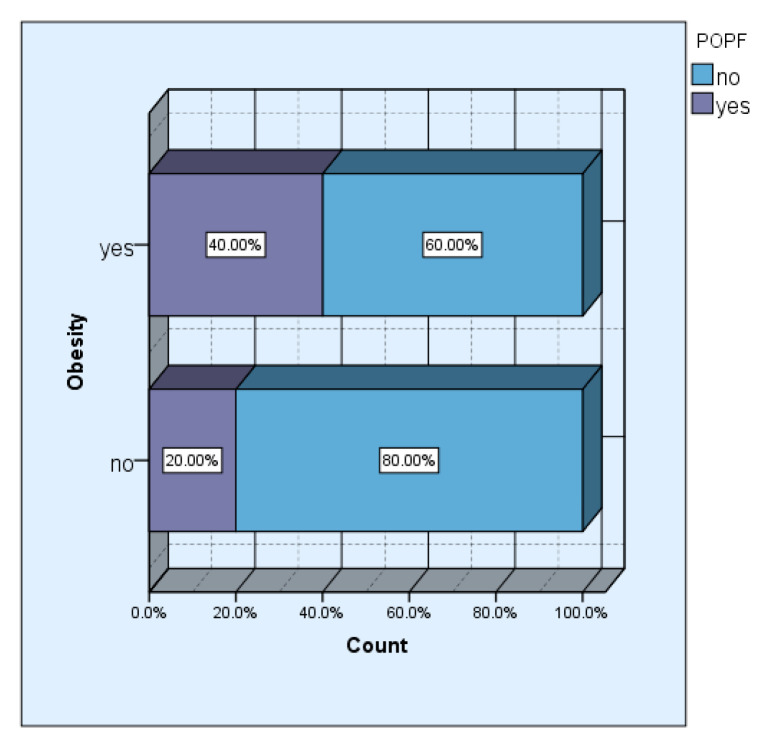
Obesity vs. POPF: in the obesity group, the incidence of POPF is 40%, and in the non-obesity groups, it is 20% (the x axis = the incidence of POPF%).

**Figure 4 diagnostics-15-00268-f004:**
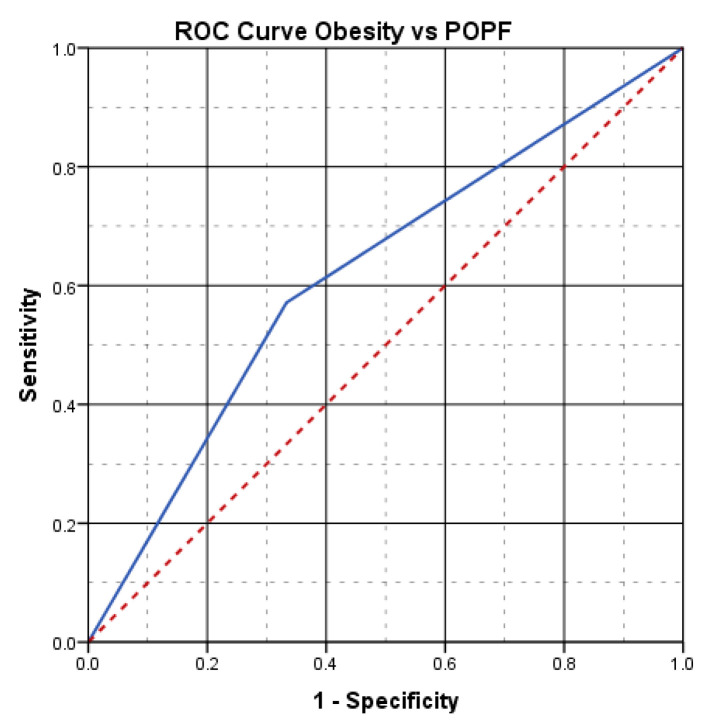
Obesity vs. POPF: ROC curve (obese patients present an increased risk for POPF).

**Figure 5 diagnostics-15-00268-f005:**
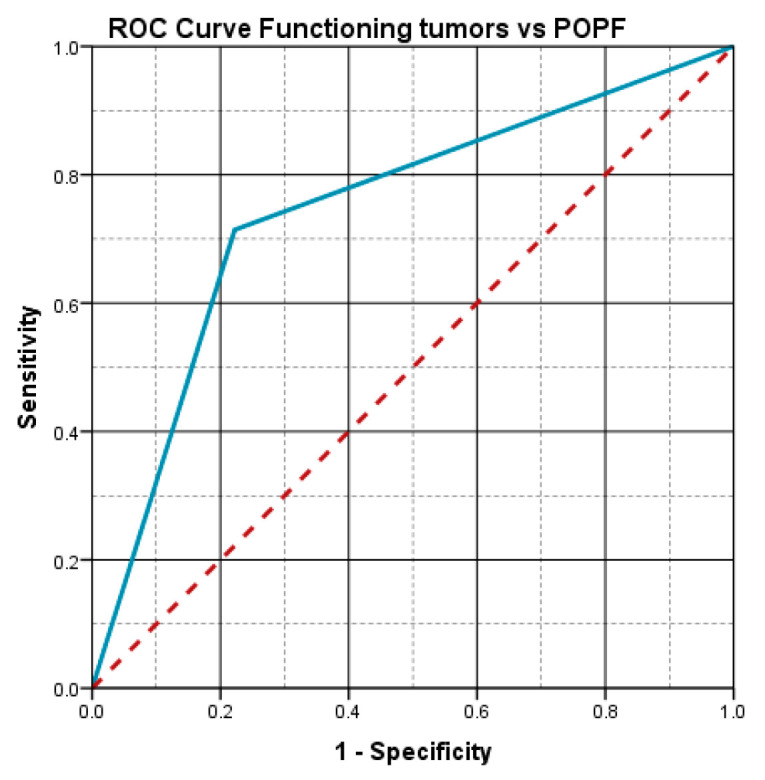
Functioning tumors vs. POPF: ROC curve.

**Figure 6 diagnostics-15-00268-f006:**
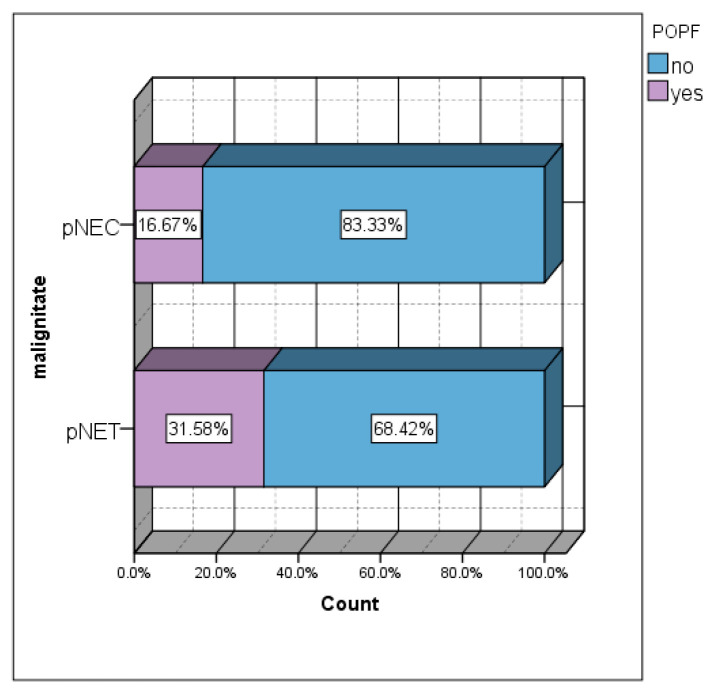
pNET/pNEC vs. POPF: (the x axis = the incidence of POPF% in the two groups: pNEC and pNET).

**Figure 7 diagnostics-15-00268-f007:**
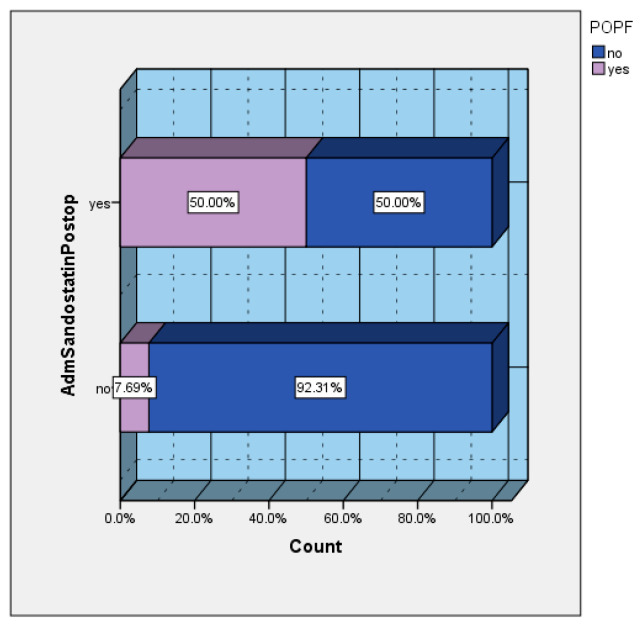
The incidence of POPF in the prophylactic sandostatin group (x-axis shows the incidence of POPF in group that received prophylaxis with sandostatin postoperatively).

**Figure 8 diagnostics-15-00268-f008:**
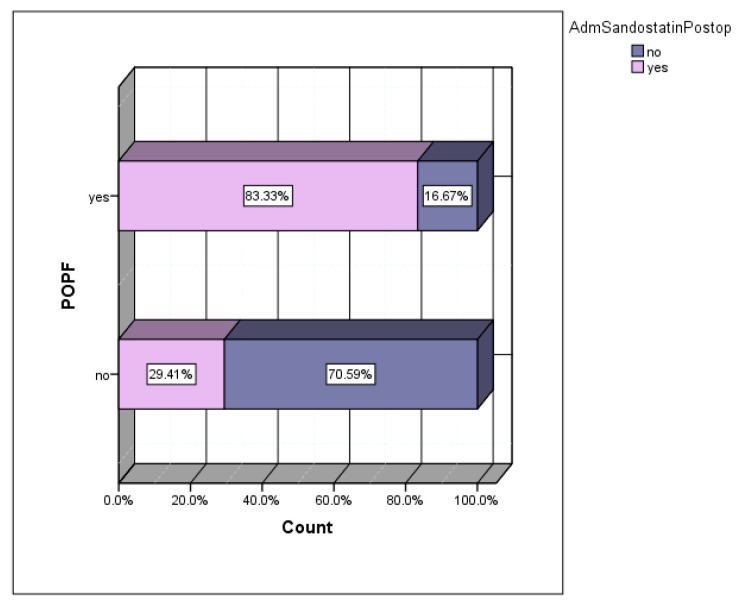
The frequency of prophylactic sandostatin in the POPF group (83.33% of patients with POPF received prophylaxis with sandostatin).

**Figure 9 diagnostics-15-00268-f009:**
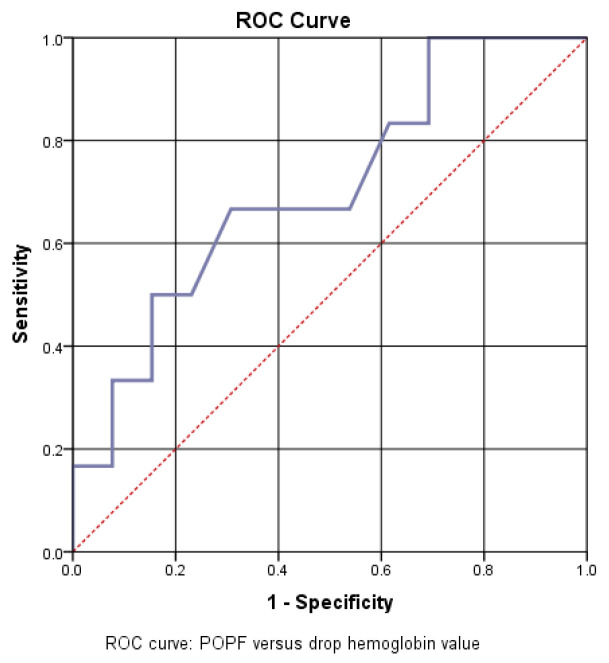
ROC curve—POPF versus drop in hemoglobin value.

**Figure 10 diagnostics-15-00268-f010:**
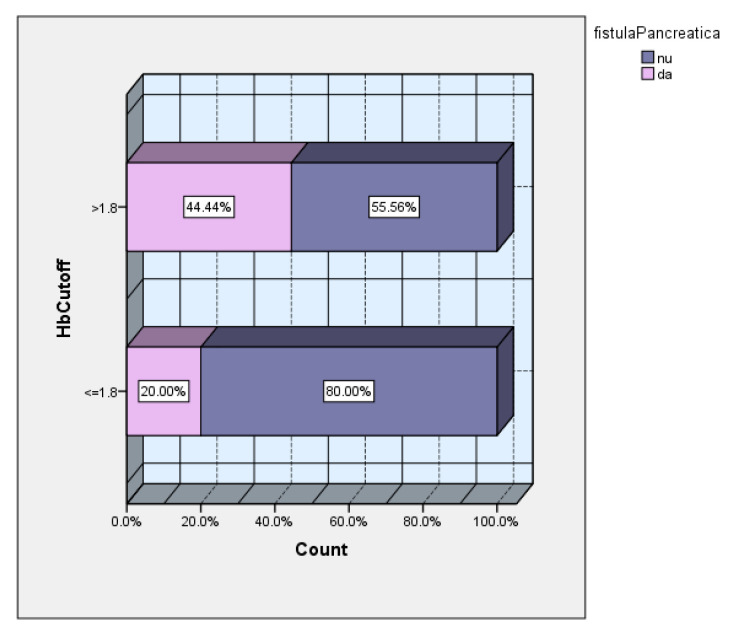
POPF vs. cutoff value of drop in hemoglobin.

**Figure 11 diagnostics-15-00268-f011:**
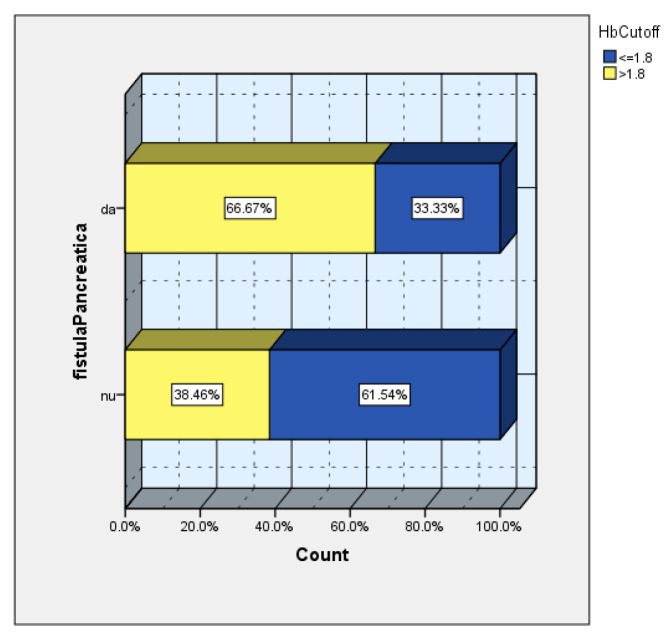
The frequency of POPF according to cutoff value (66.67% of patients with POPF had a drop in hemoglobin more than 1.8 mg/dl).

**Table 3 diagnostics-15-00268-t003:** Tumor size vs. POPF.

POPF	Tumor Size		Total
	T ≤ 2 cm	T > 2 cm	
No %within POPF	40%	60%	100%
%within Tumor	60%	75%	68.2%
Yes %within POPF	57.1%	42.9%	100%
%within Tumor	40%	25%	31.8%
Total %within POPf	45.5%	54.5%	
%within tumor	100%	100%	

**Table 4 diagnostics-15-00268-t004:** Correlations between POPF and functioning character of tumors.

Correlations				
			POPF	Functioning
Kendall’s tau_b	POPF	Correlation coefficient	1.000	0.460 *
		Sig. (1-tailed)	.	0.012
		N	25	25
	functioning	Correlation coefficient	0.460 *	1.000
		Sig. (1-tailed)	0.012	.
		N	25	26
Spearman’s rho	POPF	Correlation coefficient	1.000	0.460 *
		Sig. (1-tailed)	.	0.010
		N	25	25
	functioning	Correlation coefficient	0.460 *	1.000
		Sig. (1-tailed)	0.010	.
		N	25	26

* Correlation is significant at the 0.05 level (1-tailed).

**Table 5 diagnostics-15-00268-t005:** Correlations between POPF and prophylactic administration of sandostatin.

Correlations				
			POPF	Prophylactic Sandostatin
Kendall’s tau_b	POPF	Correlation coefficient	1.000	0.478 *
		Sig. (1-tailed)	.	0.013
		N	25	23
	Prophylactic sandostatin	Correlation coefficient	0.478 *	1.000
		Sig. (1-tailed)	0.013	.
		N	23	24
Spearman’s rho	POPF	Correlation coefficient	1.000	0.478 *
		Sig. (1-tailed)	.	0.011
		N	25	23
	Prophylactic sandostatin	Correlation coefficient	0.478 *	1.000
		Sig. (1-tailed)	0.011	.
		N	23	24

* Correlation is significant at the 0.05 level (1-tailed).

**Table 6 diagnostics-15-00268-t006:** Correlation between POPF and drop in hemoglobin.

Correlations				
			POPF	Drop in Hemoglobin
Kendall’s tau_b	POPF	Correlation coefficient	1.000	0.317 *
		Sig. (1-tailed)	.	0.044
		N	25	22
	Drop in hemoglobin	Correlation coefficient	0.317 *	1.000
		Sig. (1-tailed)	0.044	.
		N	22	22
Spearman’s rho	POPF	Correlation coefficient	1.000	0.372 *
		Sig. (1-tailed)	.	0.044
		N	25	22
	Drop in hemoglobin	Correlation coefficient	0.372 *	1.000
		Sig. (1-tailed)	0.044	.
		N	22	22

* Correlation is significant at the 0.05 level (1-tailed).

## Data Availability

The data published in this research are available on request from the first and last author and corresponding author.

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
