# Peer review of "Predicting Factor for Occurrence of Postoperative Pancreatic Fistula in Patients with Pancreatic Neuroendocrine Tumors"

_diagnostics, 2025, doi:10.3390/diagnostics15030268_

Round 1
Reviewer 1 Report
Comments and Suggestions for Authors
Summary: The authors have identified associations between incidence frequency of pancreatic fistulas and risk factors, including dynamic hemoglobin value post-resection of pancreatic neuroendocrine tumors. The study group consisted of 26 patients, and as key results, presence of obesity, functioning tumors, and hemoglobin value were shown to be associated with the risk of developing POPF. In addition, an association between POPF incidence and prophylactic sandostatin treatment was shown. ROC curve and AUC value were provided for all key findings.
Major points
· Since the title emphasizes dynamic hemoglobin value, additional data discussing the same will be valuable. Otherwise, the title should be edited to align with the present form of the article.
· Please introduce pancreatic fistula in the introduction in terms of definition and types.
· Methods like IHC should be described.
· Adding subheadings to results will help summarize the major findings within experiments.
· Fig 1. Tumor Size – What is the X axis? If it is tumor size, then what is the unit? Line 92 states that the tumor sizes vary between 9 and 65 mm. So I am assuming the x-axis is in cm. Detailed labeling of the X and Y axes is crucial for understanding the data.
· Define MEN1 syndrome.
· Can a table be made for all the information given in Lines 96-107 and Lines 117-120?
· Mention of Figure 2 in the text is missing.
· Figures 2 and 5 need a more detailed legend. Is count indicative of patient population%? Please edit the X-axis label accordingly.
· Given that ROC curves are not enough for a “good” conclusive medical statement, can other correlation coefficients be performed on the data as a validation of the ROC data?
· Line 145 – please be consistent in tumor size unit throughout the paper.
· Lines 141-147: Even if it is not significant, since the data has been mentioned in the paper and a concluding correlation statement was made in lines 145-147, please show the graph/results as supplementary material.
· Figures 4 and 8 – Can the predictive ROC curve be validated in some way? Please see the above comment on ROC curves.
· Lines 156-158. The authors state that the lowest incidence of POPF was in the patients with malignant tumors, which is 14.3%, as shown in Fig 5. Where is the 14.3% labeled in the axis? I am assuming we are looking at the Y axis. Can the graph in Figure 5 be made in any other way or labeled in detail to point out the same? The way it is now, it needs a lot of “figuring out” to understand the sentence and the corresponding graph. And also, why the focus is not on the “highest incidence of POPF”?
· Please add a reference for line 231.
· The discussion needs to be rewritten to focus on the findings of the paper. The present form of the discussion mostly looks like a review of other articles.
· What is the clinical significance of this article, and how is it different from other published literature on risk factors for POPF? If the difference lies in the identification of the dynamic hemoglobin value (a rightfully underexplored factor in POPF), the article should be rewritten or edited to focus entirely on the same.
Author Response
First of all, we, the authors, thank you for the objective assessments and recommendations that will help us improve the scientific quality of this article.
We modified the title, and hope the new one is more appropriate. We added the definition of pancreatic fistula and their classification in the text.
Regarding the IHC data, we did not put into the method how the preparation of the resection parts and the IHC stainings were carried out, because our study does not refer to the diagnosis of these tumors, but we analyzed the postoperative factors. We thought that this information is not necessary, considering that they are standard methods, just as we did not describe the surgical intervention techniques (which are standard, ex Whipple operation). However, I added an image with the histological aspect. If you consider it necessary to complete the standard IHC methods, we will add this information.
For each image I have added additional information, out of the desire to become clearer and we introduced the definition of MEN1 syndrome in the text.
We added subtitles to the obtained results, and indeed this change helped us to make the results much clearer and easier to follow.
Regarding the ROC curve and AUC, this statistical method allows us to identify the risk factors that present a good predictive power for the occurrence of a certain event, in our case of postoperative pancreatic fistula. Unfortunately, due to the small group of patients, we are limited to statistical analysis tests, because most of them are indicated for groups of over 30 patients. Thus, the results can be falsely negative (not to obtain statistical significance due to the small number of patients).
So, in this context, we have identified articles in which it is mentioned that for small groups of patients, a significant p-value up to a p=0.1 value can be accepted, precisely in order not to lose data that can be useful, and that could become powerful significant on large groups.
But, we preferred to publish only the data for which we obtained statistical significance, and we did not extend to a higher p-value, precisely so that there are no doubts about the results obtained.
Thank you for drawing our attention to the units of measure. We complied and modified, so that there is a single unit of measure.
Also, we added to the discussions information from our study, so that we could relate the results obtained by us to the data from the specialized literature.
Pancreatic neuroendocrine tumors are rare tumors. Therefore, for our university center, this study group represents a large number of patients. We wanted to carry out this study because there are not many studies that analyze the risk factors for POPF for patients with pNETs. pNETS patients have a different evolution compared to the other types of pancreatic tumors, and in this context, I think that the risk factors can also be different. The novelty could be this hemoglobin drop.
It is true that the main factor for the drop in postoperative hemoglobin is closely related to intraoperative blood loss. But there are other factors that can be equally important, such as a large amount of fluids administered intravenously both intraoperatively and perioperatively, which lead to hemodilution and implicitly to a decrease in hemoglobin. The implications of these factors and their interaction in producing hemoglobin drop will be a new direction of research for new studies, possibly this study associated with a prospective one in which to observe if a better management of fluids administered perioperatively and intraoperatively leads to a decrease in hemoglobin and of the incidence of postoperative pancreatic fistulas.
There are studies that encourage the publication of the results obtained on small groups, precisely so as not to lose certain research ideas, which could be taken up and followed in the large research centers. That is precisely why we published these data as a small study, and not just as a series of cases.
Thank you for the suggestions received.
Sincerely, the authors
Reviewer 2 Report
Comments and Suggestions for Authors
I have read with great interest the research article proposed by N. Vlad and colleagues. The argument treated by the Authors in the manuscript entitled “Dynamic hemoglobin value associated with a series of perioperative factors in predicting the postoperative pancreatic fistula after pancreatic resections for pancreatic neuroendocrine tumors” is of great interest for the scientific community and for pancreatic cancer research.
To my opinion there are many aspects that should be elucidated.
The title should be more concise without repetitions.
The Abstract should be structured according to results and conclusions.
Specifically, in the results the Authors should report the statistical analysis with the statistically significant associations. The obesity, the size of tumor and the tumor malignancy did not reach the statistical power to POPF prediction. Probably this aspect is also related to the limited cases (26 patients). Dynamic analysis of hemoglobin is the predictive factor for POPF occurrence and this is in accordance with other study that documented the relationship with intraoperative blood loss and POPF.
The Authors have reported the study of Partelli et al (line 340 page 10), with reference num. 10, but in the reference list there is the study of Marx M et al. All the references should be carefully revised with correct citations.
In the diagnosis of pancreatic cancer, why didn’t the Authors indicate the EUS-guided fine needle aspiration and fluid cytology? In this study the size of pancreatic tumors varied between 9 mm and 65 mm and only 7 patients were with functional pNETs. In addition, 8 cases underwent cephalic duodenopancreatectomies. The use of preoperative biopsy with cytology is crucial for the correct indication of neoadjuvant chemotherapy, according to current protocol.
There are minor grammatic revisions.
Thank you for your proposed manuscript.
Author Response
First of all, we, the authors, thank you for the objective assessments and recommendations that will help us improve the scientific quality of this article.
Title: I tried to make the new title more concise
I have modified the summary according to your instructions
Unfortunately, the study group is a small one, considering the incidence of these tumors among the general population, in concordance to the size of the university center where this study is conducted. The great disadvantage of studies with small groups of patients is related to the statistical analysis, because the results obtained cannot reach values ​​accepted as significant.In this context, I searched in the specialized literature what we can do when the results obtained on small groups of patients are at the limit. So we have identified articles in which it is mentioned that for small groups of patients, a significant p-value up to a p=0.1 value can be accepted, precisely in order not to lose data that can be useful, and that could become powerful significant on large groups.
That study also encourages the publication of the results obtained on small groups, precisely so as not to lose certain research ideas, which could be taken up and followed in the large research centers. That is precisely why we published these data as a small study, and not just as a series of cases.
It is true that the main factor for the drop in postoperative hemoglobin is closely related to intraoperative blood loss. But there are other factors that can be equally important, such as a large amount of fluids administered intravenously both intraoperatively and perioperatively, which lead to hemodilution and implicitly to a decrease in hemoglobin. The implications of these factors and their interaction in producing hemoglobin drop will be a new direction of research for new studies, possibly this study associated with a prospective one in which to observe if a better management of fluids administered perioperatively and intraoperatively leads to a decrease in hemoglobin and of the incidence of postoperative pancreatic fistulas.
Thank you for drawing our attention to the bibliography. It is a mistake that escaped us and is not admissible. We fixed this problem.
Regarding the EUS-guided fine needle aspiration, we confirm that it is a mandatory procedure, only that during the period in which the patients included in the study were hospitalized, in our hospital these biopsies were performed extremely rarely, because the team was in training (initially), and on the other hand, they were also limited by financial resources, as there were no funds allocated for punction needles. We currently have a young team that performs EUS-guided fine needle aspiration, biopsies, they are in full ascension, and in future studies we hope that most patients will have benefited from a complete exploration and according to medical standards.
We also assume the shortcomings regarding the English of the article, so we will opt for a professional review of the text in English by a mdpi professional.
Thank you for the suggestions received.
Sincerely, the authors
Reviewer 3 Report
Comments and Suggestions for Authors
This is a study that tries to associate dynamic hemoglobin value with a series of perioperative factors in predicting the postoperative Pancreatic Fistula The study population is patients that underwent pancreatric resections for pancreatic neuroendocrine tumors.
1. Fistly, I fail to understand what dynamic hemoglobin is. Is not a common term. Please explain.
2. Secondly I do not understant from text an important correlation with dynamic hemoglobin, yet with all other well-investigated factors of POPF.
3. An impontant detail not mentioned is the exact surgery that the patient underwent. Whipple?Periferal?
4. Please correct the references' style.
5. The definition of pancreatic fistula- biochemical leak and cr-popf should be mentioned in text.
6. When abbrevations are mentioned in figure or tables, explation is needed.
7. Figure 5, 6, 7 are difficult to understand. You may choose to present differences in another way.
Author Response
First of all, we, the authors, thank you for the objective assessments and recommendations that will help us improve the scientific quality of this article.
- It is possible that this term "hemoglobin dynamic value" is not the most suitable combination of words, but we thought that it best exemplifies the results of this study. maybe it would be more appropriate the variations of the hemoglobin value, or the drop of the hemoglobin value. When we use this expression, we mean the difference between the hemoglobin value from the first postoperative day, compared to the hemoglobin value from the days before the surgery.
- In this study, we identified a series of risk factors for the occurrence of postoperative pancreatic fistulas. All these are independent risk factors. Our small study group did not allow us to do more statistical analysis tests to highlight their interaction and their interdependence in this complex mechanism of producing this postoperative complication.
- The type of surgery was chosen depending on the size and location of the tumor. Thus, pancreaticoduodenectomies (Whipple type), distal pancreatectomies (distal pancreatic resections - body and tail, associated or not with splenectomy) were performed. Enucleation was performed for small tumors that were not in contact with the pancreatic ducts.
- I'm sorry, but I don't know exactly what type of style you're referring to. If you could indicate the style of the references, we will make the modification according to the indications.
- We added the definition of pancreatic fistula and their classification in the text
- and 7. We reviewed all the tables and figures and tried to make them as clear as possible and to explain them.
Thank you for the suggestions received.
Sincerely, the authors
Round 2
Reviewer 1 Report
Comments and Suggestions for Authors
The authors have improved the paper, which is now clearer. The authors technically have addressed some of my concerns, and I recognize the effort, but there are still a few “major” points that have remained unresolved.
My concern regarding the ROC curves: Although I appreciate the authors explaining their choice of ROC curves and AUC and p-value selection, what I suggested in my initial comments was to include/add another correlation coefficient analysis performed on the data as a validation of the already done ROC curves.
In other words, while ROC curves and AUC are commonly used to depict prediction power or association and to indicate the direction of the data, they alone are not sufficient for accurate clinical conclusions. Therefore, additional correlation analysis is needed for key findings. For example, some validation correlation tests performed on a smaller sample size will include Matthews Correlation Coefficient (MCC), Spearman's rank correlation coefficient, or Kendall's tau correlation (all recommended for smaller sample sizes). This will be shown along with or in addition to ROC curves and AUC.
Some other points that were not addressed:
· Please add a reference for line 231/line 288 in the revised version.
· A table made for all the information under “Histologically and functioning characteristics of tumors” stated in Lines 96-107 (Lines 116-130 in the revised version) would make it clearer.
· Lines 198-200 – is it corresponding to figure 5 or figure 6?
· Please address the previously made comments on Figure 6: The authors state that the lowest incidence of POPF was in the patients with malignant tumors, which is 14.3%, as shown in Fig 5. Where is the 14.3% labeled in the axis? I am assuming we are looking at the Y- axis. Can the graph in Figure 5 be made in any other way or labeled in detail to point out the same? The way it is now, it needs a lot of “figuring out” to understand the sentence and the corresponding graph. And also, why the focus is not on the “highest incidence of POPF”?
Additional points
· Please check for typos throughout the document.
· Discussion is a bit too long and could be improved by reducing the parts that extensively highlight other studies.
Author Response
Thank you for the rigorous analysis of our study. This aspect helped us to improve the study, to realize where the weak points are and will certainly help us to be much more careful in future studies.
The major change was in the statistical analysis, so we added the statistical correlation tests mentioned by you, so we obtained a statistically significant p, thus supporting the results of the ROC curve. We do not have a professional statistician in our team (we, the surgeons, learned the statistics program individually, and we still have a lot to improve our knowledge).
Regarding the bibliographic references from line 288, we have already added bibliographic references.
For the information provided on lines 116-130, I also added a small table.
Regarding the inconsistency of the percentage in the text with the one in the figure, I have corrected this mistake. It was probably a drafting mistake, which should not have existed and should not have been overlooked. For this aspect, we are grateful to you.
Best regards!
The authors
Reviewer 3 Report
Comments and Suggestions for Authors
The revised rersion of your article is clearly improved. Well done!
Author Response
Answer for reviewer 3- round 2
With much respect, the authors thank you for the suggestions that helped us to improve the scientific quality of this article. We will consider the aspects pointed out by you in future articles.
Best regards,
The authors
Round 3
Reviewer 1 Report
Comments and Suggestions for Authors
The authors have successfully addressed my concerns/suggestions.
I would only ask for a minor edit which I am assuming is a typing error:
· For Table 1: Please check the following: In the text it is stated “From the point of view of hormonal secretion, 7 patients were with functional pNETs and 19 with non-functional ones”, but table 1 shows 12 non-functioning tumors under pNETs. Please edit accordingly.
Author Response
Thank you once again for your suggestions and for your attention to our study.
I have completed the table with a column and a row each, so that it is much clearer. And with this modification I realized that a table was necessary to centralize the data and make the information clearer. I also modified the text, which really wasn't very clear.
Best regards,
The authors